# Trends in dental expenditures in Japan with a universal health insurance system

**Yukihiro Sato**[1]*, **Kakuhiro Fukai**[2], **Yuki Kunori**[1], **Eiji Yoshioka**[1], **Yasuaki Saijo**[1]

1 Department of Social Medicine, Division of Public Health and Epidemiology, Asahikawa Medical University, Asahikawa, Hokkaido, Japan, 2 Fukai Institute of Health Science, Misato, Saitama, Japan

* ys@epid.work

## Abstract

### Background

The government of Japan has spent a significant amount on dental healthcare, but it remains unknown how the spending varies across age, type of service, and time. This study describes trends in dental expenditures in Japan.

### Methods

This descriptive study used two national data sources: Estimates of National Medical Care Expenditure and Survey on Economic Conditions in Health Care. We obtained annual total and average per capita dental expenditures by age in Japan from 1984 to 2020 and estimated the proportions of types of service from 1996 to 2021. All costs were adjusted for the 2020 Consumer Price Index (1 US dollar ≈ 100 yen in 2020).

### Results

Total dental expenditures increased from 1.96 trillion yen in 1984 to 3.00 trillion yen in 2020. In particular, total and average per capita dental spending for older persons showed a rapid increase (total: from 185 billion yen in 1984 to 1.18 trillion yen in 2020; average per capita: from 15,500 yen in 1984 to 32,800 yen in 2020), contributing to the total amount increase. The crown restoration and prosthesis category amounted to 50.3% of the total expenditure in 1996, and this proportion declined to 32.4% by 2021. In 0–14 years persons, expenses on the crown restoration and prosthesis category decreased while the medical management category (mainly including fees for a management plan for oral diseases or oral functions) increased. In persons aged 65 years or older, expenses on the crown restoration and prosthesis category decreased, with increasing expenses in the medical management and at-home treatment categories.

### Conclusion

The amount of dental spending in Japan substantially increased from 1.96 trillion yen in 1984 to 3.00 trillion yen in 2020), a 1.53-fold increase. The observed changes in annual dental spending varied across age groups and types of service.

**Data Availability Statement:** The datasets of Estimates of National Medical Care Expenditure and Survey on Economic Conditions in Health Care are available in e-Stat: https://www.e-stat.go.jp/stat-search/files?page=1&toukei=00450032&tstat=

000001020931 and https://www.e-stat.go.jp/stat-search/files?page=1&toukei=00450048&tstat=000001029602.

**Funding:** This study was supported by a grant from the FUTOKU Foundation in 2022 for research projects. The funders had no role in study design, data collection and analysis, decision to publish, or preparation of the manuscript.

**Competing interests:** The authors have declared that no competing interests exist.

## Introduction

Japan has a universal health insurance system covering every resident [1,2]. Under the system, insured members receive a wide range of dental healthcare services, such as general restoration, oral surgery, and prescription, but excluding most prevention procedures [3–5]. The fee-for-service reimbursement is based on a uniform national tariff schedule, which includes the costs of dental materials for restoration. The out-of-pocket ratio for dental services is relatively low compared to other industrialised countries [4,6]. The Japanese population has been down-sizing since 2008 [7]; however, dental health expenditures in Japan have been increasing and are forecasted to maintain this trend [8]. The spending on dental healthcare in Japan is among the largest in the OECD countries [8]. Previous studies on dental expenditures in Japan have primarily focused on total costs without examining the spending pattern across different age groups, types of services, and over time [3,8]. A detailed examination of trends in dental expenditures is crucial for the development of future sustainable plans for the allocation of financial resources [9]. By understanding the specific patterns and trends in dental expenditures, policymakers can make decisions to ensure effective distribution of resources in the dental healthcare sector [9,10].

The standardised fee-for-service setting allows the government to regulate expenditures to a certain extent [2]. Since the 1980s, healthcare cost optimisation has been promoted [2,11]. The increment rate of fees for each dental service has remained relatively small and stable since 1982 [12]. In addition, the out-of-pocket payment ratio continues to increase, especially for older people. In 1973, there were no out-of-pocket fees for people aged 70 years or older. However, after 1982, they were imposed a small co-payment, and in 2001, their out-of-pocket ratio was 10% [2,11]. In 2000, the long-term care insurance system was established [2,11], with the spread of home-visit dental care [13]. In 2012, the Ministry of Health, Labour and Welfare (MHLW) set out to reform 'treatment-centred' dentistry to 'treatment, management, and interprofessional collaboration' dentistry in the health insurance system [14]. This long-term aim strengthened the role of family dentists. In 2016, certification for dental clinics providing sufficient family dentist functions was established, and certified dental offices could impose additional fees for medical management [15]. The Japanese government has been reforming the health insurance system in response to changes in economic and demographic situations. Examining whether such reforms potentially affect the financial distribution of dental spending can provide policymakers with basic knowledge about future resource planning.

Although the government controls healthcare costs through a review of fees [2], the natural distribution of oral diseases and age structure of populations are implicated as factors that can change dental healthcare spending [10,16–18]. Overall, the oral health status in Japan has been improving with better oral hygiene behaviours [3,19]. However, the number of older people in Japan has increased (from 12.5 million in 1985 to 36.0 million in 2020) [7], and the proportion of older persons who had 20 teeth or more has also increased (31.4% in 1993 to 73.0% in 2016 among older aged 65–69 years) [3,19]. As a result of the increased number of teeth, older individuals tend to have decayed teeth and 4 mm over of periodontal pocket depth [3,19]. Therefore, while the demand for prosthetic treatments, such as dentures, could have decreased, the demand for treatments for dental caries and periodontal disease could have increased. On the other hand, the number of children aged 15 or lower has reduced (from 26.0 million children aged 15 years or less in 1985 to 15.0 million in 2020) [7]. In addition, the DMFT index of 12 years old has been rapidly declining (4.63 in 1985 to 0.68 in 2020) [3,19]. However, the decrease in the number of children per family may have led to an increase in the per capita dental healthcare costs for maintaining better oral hygiene. Each age group has its own distinct characteristics of oral disease distribution, which affects the distribution of dental

expenditures. It is necessary to describe the amount of dental spending according to the age group and type of service.

There are raised concerns about the impact of the rising prices of precious dental metals. In Japan, gold-silver-palladium (GSP) alloys are often used for restorative treatments [20]. Our previous study reported that the number of insurance claims for dental GSP alloys decreased, possibly because of the rising costs [21]. However, the sudden price increase of dental GSP alloys may have created an economic burden on dental expenditures. The MHLW has introduced new materials for insurance coverage, such as hard resin jacket crowns and computer-aided design/computer-aided manufacturing (CAD/CAM) crowns, in response to the price increase in precious metals. Thus, the application of newer technologies and materials is recognised as a factor in changing healthcare spending [2]. However, it is still unclear whether these innovations have successfully replaced precious metal materials with low-cost materials and further reduced the financial burden of the dental healthcare budget.

This study aims to firstly outline the changes in total dental expenditures according to age groups and types of service and secondly, to estimate the amount of the cost of metal materials for dental restoration.

## Methods

### Data sources

This descriptive study used two data sources. One was Estimates of National Medical Care Expenditure (ENMCE) [22], and the other was Survey on Economic Conditions in Health Care (SECHC) [23].

From ENMCE, we obtained the annual total and average per capita dental spending from 1984 to 2020. The amount covers all costs of dental insurance services for all ages and comprises the health insurance system of public expenses, social insurance, and out-of-pocket expenses. Yen amounts were adjusted for the 2020 Consumer Price Index to account for inflation over the study period for comparability [24]. In 2020, one US dollar was approximately equivalent to 100 yen [25].

We calculated proportions for each type of service in the total dental expenditure using the SECHC data because detailed expense data according to a type of service were not available from ENMCE. SECHC reported the expenditures of each dental procedure category for every year from 1996 to 2021, but these datasets consisted of information for June and not the year. The MHLW classified dental procedures into 15 areas: initial- and repeat-consultation fee (A), medical management, mainly including fees for a management plan for oral diseases or oral functions (B), at-home treatment, mainly including visit fees for at-home care, but fees for treatments and other services are not included (C), tests, mainly including endodontic, periodontic, occlusal function tests (D), diagnostic imaging, mainly including radiography and the diagnoses (E), drug administration (F), injection, mainly for cancer treatments (G), rehabilitation, mainly relating to dentures and dysphagia (H), treatment, mainly including root canal treatment and periodontal treatments (I), surgery, mainly including tooth extraction, replantation, gingivectomy, and surgeries for oral cancer (J), anaesthesia, mainly including local and general anesthesia (K), radiotherapy (L), crown restoration and prosthesis, mainly including crown and prosthetic restoration and the cost of restoration materials (M), orthodontic treatment, restricting to patients with specified diseases such as orofacial clefts (N), pathological diagnosis (O), hospitalisation fee, and others. When the pathological diagnosis category was introduced in April 2008, the others category was deleted.

Additionally, we calculated the proportion of dental material fees in the total dental expenditures from the crown restoration and prosthesis (M) category from the SECHC data. We

classified dental material fees into four categories: dental GSP alloys, dental non-metal materials, other dental metal materials, and others. The spending on dental GSP alloys includes all the costs of dental GSP alloys for crown restoration and prostheses. Dental non-metal materials included the material costs, such as filling materials, hard resin jacket crowns, CAD/CAM crowns, and resin denture bases. Other dental metal materials included the cost of gold alloys, silver alloys, nickel-chromium alloys, amalgam, cobalt-chrome alloys, titanium alloys, and stainless steel for crown restoration and prostheses. Others included the cost of minor materials, such as denture soft reline materials, reinforcement wire, and adhesive dental materials for restoration.

We estimated the expenditure amount on each dental service and dental material category by multiplying their proportions by the total dental spending. All dental expenditures were computed and graphed across various groups using the R software (version 4.1.2; R Foundation for Statistical Computing, Vienna, Austria) on macOS.

### Ethical approval

This study used publicly available datasets obtained from official national surveys conducted by the government of Japan. The datasets do not contain any personal information; therefore, ethical approval was not required.

## Results

Table 1 and Fig 1 represent the trends in inflation-adjusted total and average per capita dental expenditure. Total dental expenditures increased from 1.96 trillion yen (approximately 19.6 billion US dollars) in 1984 to 3.00 trillion yen in 2020 (approximately 30.0 billion US dollars), a 1.53-fold increase. Dental expenditures of individuals aged 0–14 years and 15–44 years decreased from 264 billion yen and 915 billion yen in 1984 to 250 billion yen and 719 billion yen in 2020, respectively, although the trends fluctuated during the study period. On the other hand, dental expenditures of those 45–64 years and 65 years or older increased from 591 billion yen and 185 billion yen in 1984 to 850 billion yen and 1.18 trillion yen in 2020, respectively. The average per capita spending was JPY 16,300 in 1984, which increased to JPY 23,800 in 2020. Excluding those 15–44 years, the average per capita spending of all the other age groups increased from 1984 to 2020. In particular, a rapid increase in the average per capita spending of those 65 years or older was observed across the study period (from JPY 15,500 in 1984 to JPY 32,800 in 2020).

Fig 2 and S1–S5 Tables show the amounts and proportions of each dental service category. In 2020, the crown restoration and prosthesis (M) category had the largest amount and proportion. However, both the amount and proportion of the crown restoration and prosthesis (M) category decreased during the study period (amount: from 1.33 trillion yen in 1996 to 1.05 trillion yen in 2020; proportion: from 50.3% in 1996 to 32.4% in 2021). The treatment (I) category shows the next largest amount and proportion in 2020. The spending on the treatment (I) category increased from 496 billion yen (18.7%) in 1997 to 598 billion yen (19.9%) in 2020, but the trend was not consistent. The initial- and repeat-consultation fee (A) and the medical management (B), with the third largest amount and proportion in 2020, increased from 286 billion yen (10.8%) and 126 billion yen (4.8%) in 1997 to 384 billion yen (12.8%) and 384 billion yen (12.8%) in 2020. The spending on at-home treatment (C) categories increased by 9.9-times from 1996 to 2020, respectively.

In the groups of 0–14 years, 15–44 years, and 44–64 years old, the proportion of the medical management (B) category increased, while the crown restoration and prosthesis (M) category decreased. In particular, this trend was remarkable in those 0–14 years (B category: 3.3% in

**Table 1. Total and average per capita dental expenditures in Japan.**

| Year | Total dental expenditures | Total dental expenditures per capita | Dental expenditures in 0–14 years old | Dental expenditures per capita in 0–14 years old | Dental expenditures in 15–44 years old | Dental expenditures per capita in 15–44 years old | Dental expenditures in 45–64 years old | Dental expenditures per capita in 45–64 years old | Dental expenditures in 65+ years old | Dental expenditures per capita in 65+ years old |
|---|---|---|---|---|---|---|---|---|---|---|
| | (1 trillion yen [≈ 10 billion US dollars]) | (1000 yen [≈ 10 US dollars]) | (1 trillion yen [≈ 10 billion US dollars]) | (1000 yen [≈ 10 US dollars]) | (1 trillion yen [≈ 10 billion US dollars]) | (1000 yen [≈ 10 US dollars]) | (1 trillion yen [≈ 10 billion US dollars]) | (1000 yen [≈ 10 US dollars]) | (1 trillion yen [≈ 10 billion US dollars]) | (1000 yen [≈ 10 US dollars]) |
| 1984 | 1.955 | 16.3 | 0.264 | 10.0 | 0.915 | 17.0 | 0.591 | 21.0 | 0.185 | 15.5 |
| 1985 | 2.002 | 16.6 | 0.284 | 10.9 | 0.893 | 16.6 | 0.615 | 21.5 | 0.210 | 16.8 |
| 1986 | 2.135 | 17.6 | 0.286 | 11.3 | 0.923 | 17.1 | 0.689 | 23.5 | 0.236 | 18.4 |
| 1987 | 2.210 | 18.1 | 0.288 | 11.6 | 0.944 | 17.4 | 0.726 | 24.1 | 0.252 | 19.0 |
| 1988 | 2.267 | 18.5 | 0.289 | 12.0 | 0.962 | 17.8 | 0.746 | 24.2 | 0.270 | 19.4 |
| 1989 | 2.257 | 18.3 | 0.267 | 11.5 | 0.932 | 17.1 | 0.770 | 24.5 | 0.288 | 20.1 |
| 1990 | 2.272 | 18.4 | 0.280 | 12.4 | 0.909 | 16.7 | 0.766 | 24.2 | 0.316 | 21.2 |
| 1991 | 2.288 | 18.5 | 0.273 | 12.4 | 0.918 | 16.7 | 0.773 | 24.4 | 0.324 | 20.8 |
| 1992 | 2.441 | 19.7 | 0.275 | 12.9 | 0.972 | 17.9 | 0.828 | 25.5 | 0.366 | 22.5 |
| 1993 | 2.427 | 19.5 | 0.249 | 11.9 | 0.944 | 17.6 | 0.855 | 25.7 | 0.379 | 22.4 |
| 1994 | 2.450 | 19.6 | 0.231 | 11.4 | 0.901 | 17.1 | 0.890 | 26.0 | 0.429 | 24.4 |
| 1995 | 2.486 | 19.8 | 0.211 | 10.5 | 0.885 | 16.9 | 0.923 | 26.4 | 0.466 | 25.5 |
| 1996 | 2.649 | 21.0 | 0.206 | 10.4 | 0.918 | 17.7 | 1.004 | 28.3 | 0.521 | 27.4 |
| 1997 | 2.594 | 20.6 | 0.181 | 9.3 | 0.884 | 17.2 | 0.991 | 27.8 | 0.539 | 27.2 |
| 1998 | 2.563 | 20.2 | 0.203 | 10.7 | 0.839 | 16.4 | 0.950 | 26.6 | 0.571 | 27.9 |
| 1999 | 2.596 | 20.5 | 0.206 | 10.9 | 0.792 | 15.5 | 0.960 | 26.8 | 0.638 | 30.1 |
| 2000 | 2.628 | 20.7 | 0.212 | 11.4 | 0.781 | 15.5 | 0.972 | 27.1 | 0.663 | 30.1 |
| 2001 | 2.693 | 21.2 | 0.202 | 11.1 | 0.803 | 15.9 | 0.978 | 27.3 | 0.710 | 31.0 |
| 2002 | 2.701 | 21.2 | 0.213 | 11.8 | 0.848 | 16.9 | 0.931 | 26.2 | 0.709 | 30.0 |
| 2003 | 2.657 | 20.8 | 0.200 | 11.2 | 0.798 | 15.9 | 0.941 | 26.6 | 0.717 | 29.5 |
| 2004 | 2.657 | 20.8 | 0.195 | 11.0 | 0.788 | 15.9 | 0.907 | 25.5 | 0.768 | 30.9 |
| 2005 | 2.707 | 21.2 | 0.208 | 11.9 | 0.781 | 16.0 | 0.913 | 25.7 | 0.805 | 31.3 |
| 2006 | 2.622 | 20.5 | 0.192 | 11.0 | 0.707 | 14.6 | 0.902 | 25.8 | 0.820 | 30.8 |
| 2007 | 2.617 | 20.5 | 0.203 | 11.7 | 0.709 | 14.7 | 0.881 | 25.4 | 0.824 | 30.1 |
| 2008 | 2.663 | 20.9 | 0.204 | 11.9 | 0.731 | 15.3 | 0.855 | 24.9 | 0.873 | 30.9 |
| 2009 | 2.679 | 21.0 | 0.207 | 12.1 | 0.725 | 15.3 | 0.843 | 24.7 | 0.904 | 31.2 |
| 2010 | 2.745 | 21.4 | 0.220 | 13.1 | 0.726 | 15.4 | 0.854 | 24.7 | 0.946 | 32.1 |
| 2011 | 2.831 | 22.1 | 0.225 | 13.4 | 0.744 | 15.9 | 0.872 | 25.3 | 0.990 | 33.2 |
| 2012 | 2.871 | 22.5 | 0.230 | 13.9 | 0.742 | 16.1 | 0.862 | 25.3 | 1.037 | 33.7 |
| 2013 | 2.884 | 22.7 | 0.230 | 14.0 | 0.730 | 16.0 | 0.837 | 25.0 | 1.087 | 34.0 |
| 2014 | 2.862 | 22.6 | 0.228 | 14.1 | 0.721 | 16.1 | 0.813 | 24.6 | 1.099 | 33.3 |
| 2015 | 2.881 | 22.7 | 0.230 | 14.5 | 0.717 | 16.2 | 0.807 | 24.5 | 1.127 | 33.3 |
| 2016 | 2.913 | 22.9 | 0.239 | 15.2 | 0.714 | 16.3 | 0.811 | 24.7 | 1.148 | 33.2 |
| 2017 | 2.941 | 23.2 | 0.244 | 15.6 | 0.712 | 16.5 | 0.819 | 24.8 | 1.167 | 33.2 |
| 2018 | 2.973 | 23.5 | 0.251 | 16.3 | 0.701 | 16.6 | 0.826 | 24.8 | 1.195 | 33.6 |
| 2019 | 3.015 | 23.9 | 0.254 | 16.7 | 0.697 | 16.8 | 0.847 | 25.2 | 1.218 | 33.9 |
| 2020 | 3.002 | 23.8 | 0.250 | 16.7 | 0.719 | 17.5 | 0.850 | 25.0 | 1.183 | 32.8 |

1996 to 22.6% in 2021; M category: 39.9% in 1996 to 18.7% in 2021). In those 65 years or older, the proportion of the crown restoration and prosthesis (M) category decreased from 61.7% in 1996 to 36.9% in 2021. In contrast, the medical management (B), at-home treatment (C),

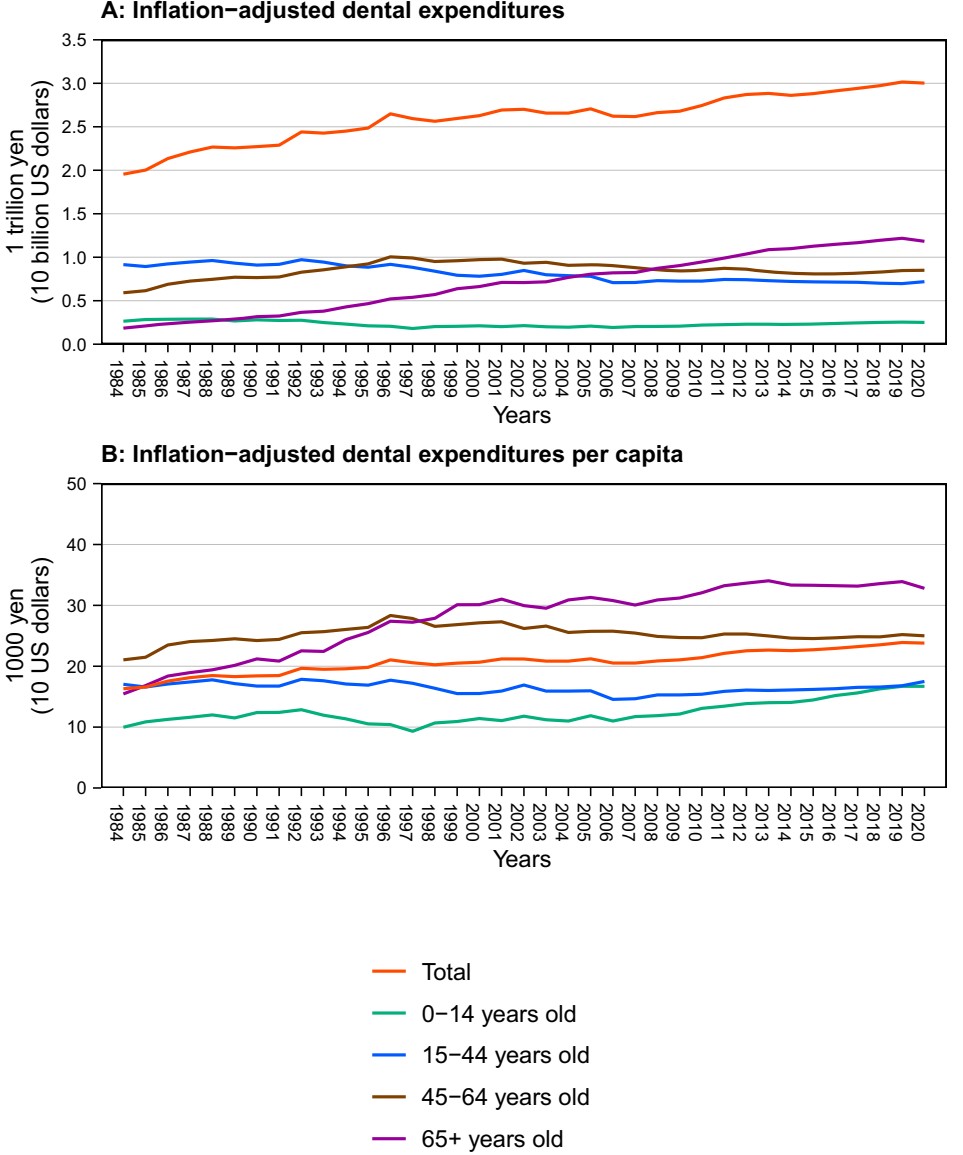

**Fig 1. Trends in the total and average per capita dental expenditures in Japan.**

rehabilitation (H), and treatment (I) categories increased. However, the total amount of the crown restoration and prosthesis (M) category for those over 65 years increased from 321 billion yen in 1996 to 449 billion yen in 2020.

**Fig 3** and **S6 Table** shows the amount and proportion of dental materials for restoration. The total amount of spending on dental materials increased from 163 billion yen in 1996 to 228 billion yen in 2020, but the trend was inconsistent. The proportion of dental materials in total dental expenditures fluctuated between 4.5% and 7.9% across the study period. The spending increased as a proportion of the crown restoration and prosthesis (M) category (from 12.3% in 1996 to 24.3% in 2021). The spending on dental gold-silver-palladium alloys increased from 94.2 billion yen (3.6% in total) in 1996 to 173 billion yen (5.8%) in 2020, accounting for 6.1% of total dental expenditures in 2021. The spending on dental non-metal

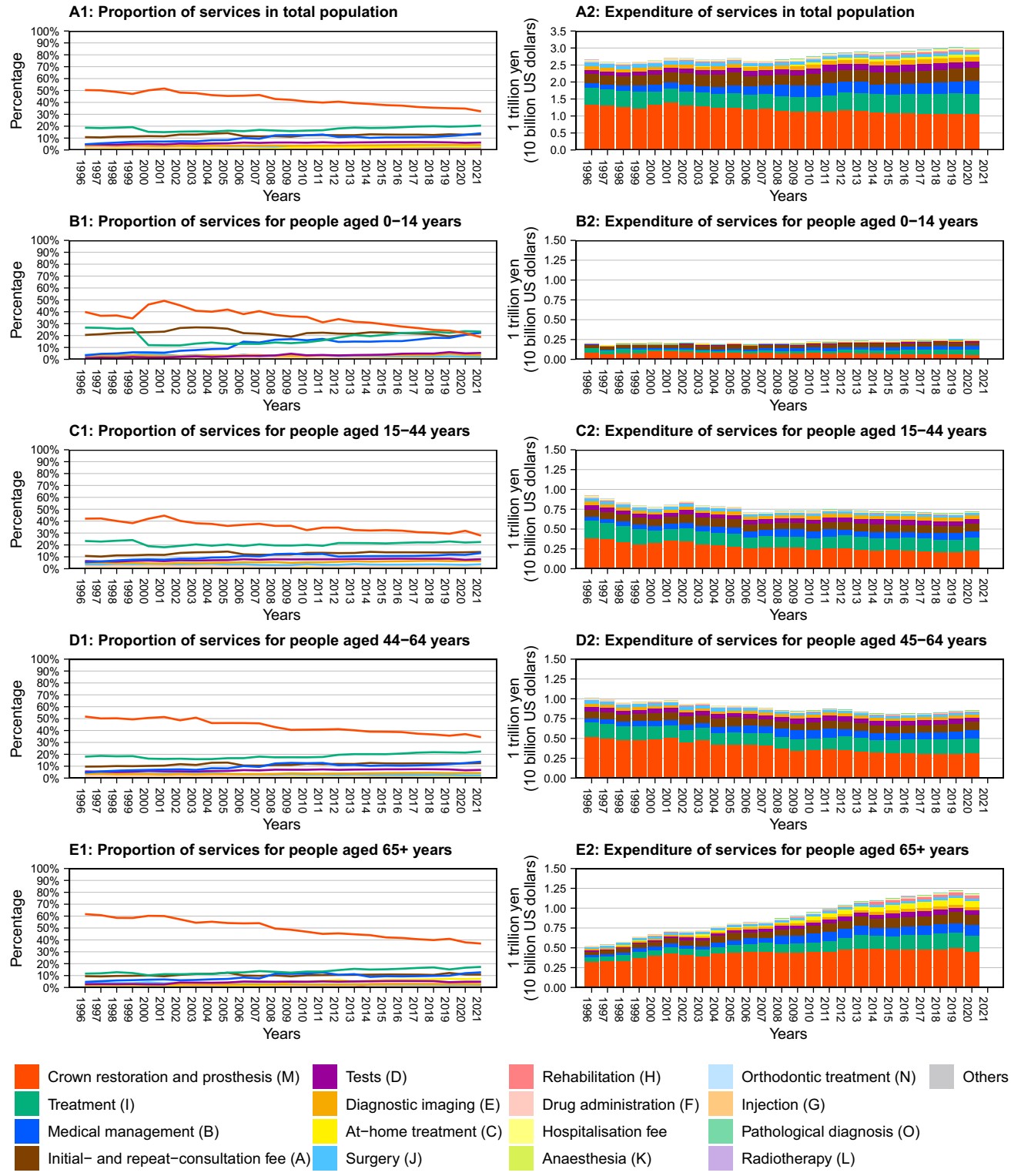

**Fig 2. Trends in the amount and proportion of services in Japan.**

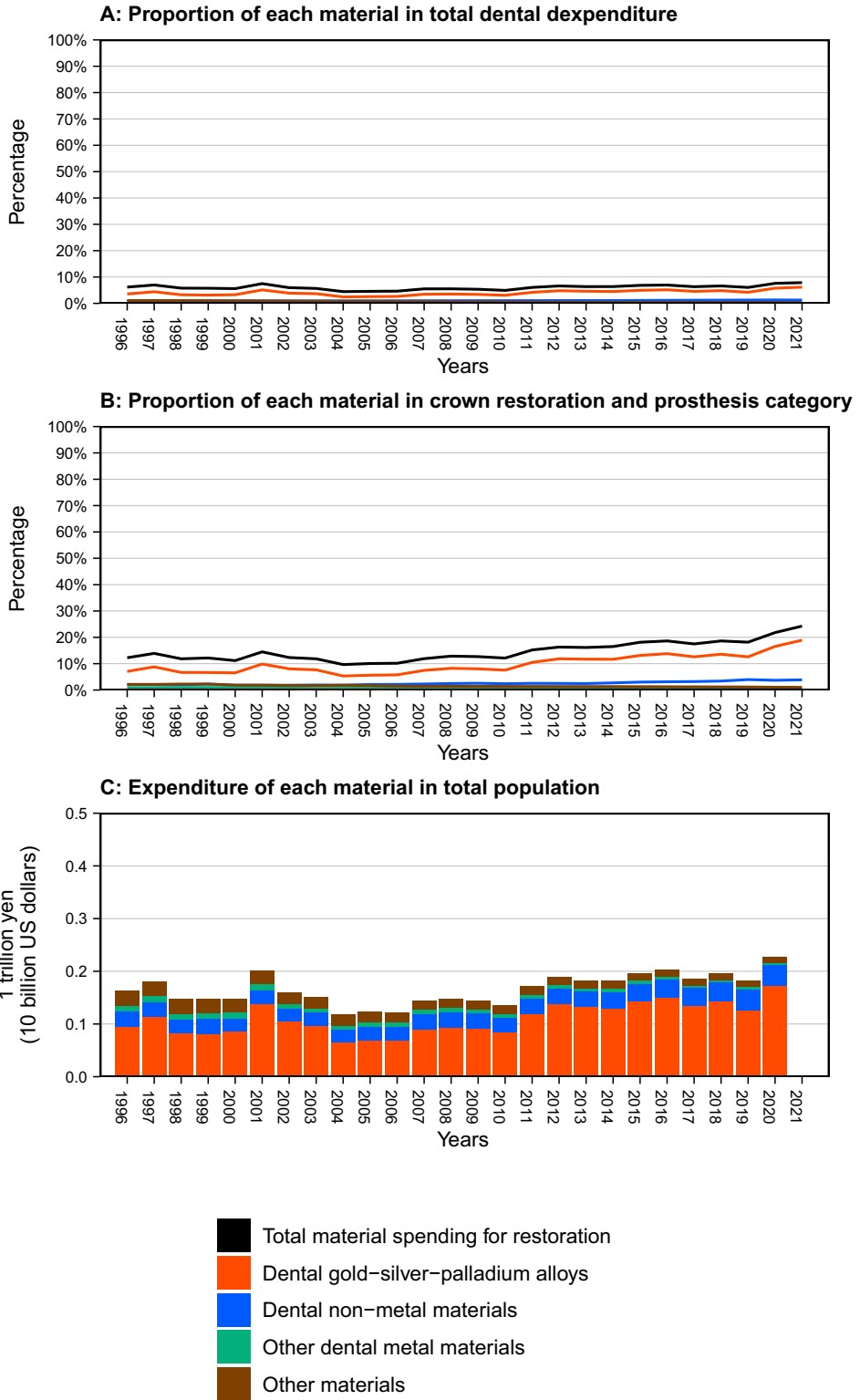

**Fig 3. Trends in the amount and proportion of dental materials for restoration in Japan.**

materials increased from 28.5 billion yen (1.1% in total) in 1996 to 38.6 billion yen in 2020 (1.3% in total).

## Discussion

This study provides new insights into dental expenditures in Japan from a longitudinal perspective. Using national databases, we reported that the total dental expenditure in Japan increased by 1.53-folds from 1984 to 2020 after adjustments for price inflation, while the government exercised control of healthcare costs by review of fees. The change in dental expenditure was non-uniform across age groups and types of service. In particular, older individuals contributed to its rapid increase. The crown restoration and prosthesis (M) category decreased in amount and proportion; however, this category had the largest amount and proportion over the study period. The total spending on dental materials for restoration was relatively stable during the study period, whereas a previous study reported the rising price of dental GSP alloys [21].

Before interpreting the results, the following limitations should be noted. First, the datasets of SECHC were June data and not annual data [23]. Therefore, the proportions of each type of dental service have random errors. Furthermore, as the sampling method has changed since 2015 [23], the data before 2014 have potentially increased the random errors. Moreover, this study did not account for underlying secular trends and extraneous factors. Therefore, these limitations can violate internal validity. Second, dental service categories do not correspond to disease-specific classifications [23]. Some categories include procedures for multiple oral diseases. Therefore, it was not possible to clearly describe the amount spent on a specific oral disease. This limitation complicates the interpretation of results. Third, we could not collect data on the number of procedures corresponding to specific oral diseases owing to the system's complexity. Therefore, it was unclear whether the total amount increased because of an increase in the frequency or fee for each procedure. However, excluding the cost of dental metal materials, as fees for each dental service have remained relatively small and stable since 1982 [12], it can be expected that the increase in the number of insurance claims for each procedure would have had a dominant impact on the increase in total dental expenditures.

Dental healthcare costs for older individuals increased rapidly, which markedly contributed to the increase in total dental expenditures. This rapid increase can be due to the growing elderly population and increased per capita costs. In particular, the number of patients aged 65 years or older who received dental services was growing [26], probably because of improving oral health status and spreading home-visit dental care. In Japan, the medical expenditures of older people have been rising, placing a heavy burden on government finances [1,11,22,27]. In particular, Japan established a late-stage medical care system for the elderly in 2008, a public health insurance system for people aged 75 years and older [1,2,11]. The insured contributes to only approximately 10% of the total cost causing the government to gradually increase their out-of-pocket payment ratio [1,2,11]. As dental spending partially contributes to the financial burden, it is important to continue monitoring trends in the distribution of dental spending and oral diseases in older people.

In total dental expenditures, the proportion of the crown restoration and prosthesis (M) category decreased. This result can reflect the situation that dental caries and tooth loss have been decreasing in Japan because of better oral hygiene behaviours [3]. On the other hand, the proportion of the medical management (B) category increased. The policy of the MHLW has emphasised the importance of continuous management of oral diseases in the last decade [14,15], which is likely to be reflected in the increase in spending on medical management. However, it is noteworthy that crown restoration and prosthesis (M) still had the largest

proportion and amount. Considering the current result, the distribution of dental expenditures seems to have been still 'treatment-centred' instead of 'treatment, management, and interprofessional collaboration'.

In older, the proportion of the crown restoration and prosthesis (M) category downsized from 61.7% in 1994 to 36.9% in 2021. In contrast, the medical management (B) and at-home treatment (C) categories were upsized, to which policy induction might have contributed. The number of older people who have received certification for needed long-term care or support has been increasing [11]. In addition, there is a high demand for oral care for older people with long-term care needs; however, dentists do not seem to provide home care sufficiently [28]. The current results highlight the need to monitor how spending on home-visit dental care will increase and whether family dentists maintain their oral functions well.

We found an increase in the total dental expenditures for 0–14 years old, although the population has been downsizing. This result can be due to an increase in spending on the average per capita. In 2002, the out-of-pocket payment ratio for children under 3 years old declined from 30% to 20%, and in 2008, pre-school-age children (up to 6 years old) were also applied [2]. Prefectures and municipalities independently provide additional medical subsidies for children under various conditions [29]. Out-of-pocket expense is among the strongest barrier to dental care utilisation by patients [30–32]. Therefore, removing financial barriers may have facilitated the usage of dental services for children. In light of the types of service, the proportion of the crown restoration and prosthesis (M) category decreased, with dental caries decreasing. On the other hand, the spending on the medical management (B) category markedly increased. The financial distribution of dental spending on children seems to shift from treatment-centred to management-centred dentistry.

Our previous study indicated that dental GSP alloys for inlays and crowns were replaced with cheaper non-metal dental materials as the price of dental GSP alloys has been increasing [21]. The spending on dental materials shows a relative increase in the crown restoration and prosthesis (M) category. This result suggests that there is an imbalance between procedure fees and material fees in the crown restoration and prosthesis (M) category owing to the rising prices of precious metals. However, the total expenditure on dental materials for restoration seems to be relatively stable during the study period. The introduction policy might have succeeded in suppressing the increase in the cost of dental materials for restoration.

## Conclusions

The amount of dental spending in Japan increased by 1.53-folds from 1984 to 2020 after adjustments for price inflation, whereas the government controls healthcare costs by review of fees. The increment in dental expenditures for older people was remarkable and contributed mainly to the total amount increase. The crown restoration and prosthesis (M) category showed lowered amounts and proportions; however, the category still had the largest amount and proportion over the study period. The spending on dental materials was found to be relatively stable, whereas the price of precious metals has been on the rise. These results could provide basic information towards efforts to further manage dental expenditures in Japan for patients, physicians, and policymakers.

## Supporting information

**S1 Table. Amount and proportion of services per year.**
(DOCX)

**S2 Table. Amount and proportion of services per year for people aged 0–14 years.** (DOCX)

**S3 Table. Amount and proportion of services per year for people aged 15–44 years.** (DOCX)

**S4 Table. Amount and proportion of services per year for people aged 45–64 years.** (DOCX)

**S5 Table. Amount and proportion of services per year for people aged 65 years or older.** (DOCX)

**S6 Table. Amount and proportion of dental materials for restoration per year.** (DOCX)

## Acknowledgments

We would like to thank the Ministry of Health, Labour and Welfare for public access to data.

## Author Contributions

**Conceptualization:** Yukihiro Sato.

**Data curation:** Yukihiro Sato.

**Formal analysis:** Yukihiro Sato.

**Funding acquisition:** Yukihiro Sato.

**Methodology:** Yukihiro Sato, Kakuhiro Fukai, Yuki Kunori, Eiji Yoshioka, Yasuaki Saijo.

**Resources:** Yukihiro Sato.

**Supervision:** Kakuhiro Fukai, Yasuaki Saijo.

**Validation:** Yukihiro Sato.

**Visualization:** Yukihiro Sato, Eiji Yoshioka.

**Writing – original draft:** Yukihiro Sato, Yasuaki Saijo.

**Writing – review & editing:** Kakuhiro Fukai, Yuki Kunori, Eiji Yoshioka.

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
