## [Decision Letter · Decision Letter 0]

2 May 2023

PONE-D-22-32885Trends in dental expenditures in Japan with a universal health insurance systemPLOS ONE

Dear Dr. Sato,

Thank you for submitting your manuscript to PLOS ONE. After careful consideration, we feel that it has merit but does not fully meet PLOS ONE’s publication criteria as it currently stands. Therefore, we invite you to submit a revised version of the manuscript that addresses the points raised during the review process.

We look forward to receiving your revised manuscript.

Kind regards,

Ahmed Jamleh

Academic Editor

PLOS ONE

2. Please ensure that you have specified (1) whether consent was informed and (2) what type you obtained (for instance, written or verbal, and if verbal, how it was documented and witnessed). If your study included minors, state whether you obtained consent from parents or guardians. If the need for consent was waived by the ethics committee, please include this information.

“This study was supported by a grant from the FUTOKU Foundation in 2022 for research projects.”

Additional Editor Comments:

Thank you for submitting your manuscript.

It is well written and comprehensive.

Kindly respond to the reviewer's comments.

Moreover, please mind the following comments:

1- Rearrange the sequence of references. In P3, I see reference no.25 comes after citing the first 9 references.

2- P6: You stated that dental procedures are classified in 14 areas but based on what you wrote (A-O), they are supposed to be 15! Please check.

3- In the reference list, kindly update the access date after checking their availability.

Reviewers' comments:

Reviewer's Responses to Questions

**Comments to the Author**

1. Is the manuscript technically sound, and do the data support the conclusions?

Reviewer #1: Yes

2. Has the statistical analysis been performed appropriately and rigorously? 

Reviewer #1: No

3. Have the authors made all data underlying the findings in their manuscript fully available?

Reviewer #1: No

4. Is the manuscript presented in an intelligible fashion and written in standard English?

Reviewer #1: Yes

5. Review Comments to the Author

Reviewer #1: 

Dear Author,

Thank you for submitting to PLOS ONE. The following points have been noted for your appropriate address :

1. Abstract: medical management category should be explicit.

2. Abstract: The conclusion section should have the main results in quantitative statements as well. 

3. Page 4, Line 58: You have stated the proportion of older persons who had 20 teeth or more in 1993 and 2016. It is too old. Authors need to report the latest data.

4. This study design did not control or adjust for underlying secular trends, so it may have failed to adequately capture dental expenditure trends. Moreover, it was not able to minimize threats to internal validity, including random fluctuations, history, and maturation; these extraneous events and changes over time.   

5. Your references and literature search shows that studies have been conducted on the trend in dental expenditures under the universal health insurance system in Japan. Please highlight what additional value your study adds and the justification for your study.

6. PLOS authors have the option to publish the peer review history of their article (what does this mean?). If published, this will include your full peer review and any attached files.

Reviewer #1: No

---

## [Author Response · Author response to Decision Letter 0]

23 May 2023

RESPONSES TO REVIEWERS’ COMMENTS

Dear Ahmed Jamleh, Academic Editor, and the reviewer

Thank you for giving me the opportunity to submit a revised draft of our manuscript. We appreciate the time and effort that the editor and reviewer have invested in offering valuable feedback. We have addressed each comment by providing point-by-point responses and have made corresponding changes in the manuscript.

Journal requirements

Thank you for checking the format of our manuscript. Following your comments, we have corrected all files to conform to the PLOS ONE format. Please see below.

and

RESPONSE

We have modified our manuscript files to adhere to the style requirements of PLOS ONE.

2. Please ensure that you have specified (1) whether consent was informed and (2) what type you obtained (for instance, written or verbal, and if verbal, how it was documented and witnessed). If your study included minors, state whether you obtained consent from parents or guardians. If the need for consent was waived by the ethics committee, please include this information.

RESPONSE

The datasets used in this study are obtained from official national surveys conducted by the government, and these datasets are publicly available. Additionally, the datasets do not contain any personal information, so we believe that the ethical concerns you mentioned are not applicable to our study. We have added the explanation in our manuscript that we used publicly available secondary datasets provided by the government.

(Line 135–138, page 7)

Ethical Approval

This study used publicly available datasets obtained from official national surveys conducted by the government of Japan. The datasets do not contain any personal information; therefore, ethical approval was not required.

“This study was supported by a grant from the FUTOKU Foundation in 2022 for research projects.”

RESPONSE

The funder has no role; therefore, we have stated that "The funders had no role in study design, data collection and analysis, decision to publish, or preparation of the manuscript." in the cover letter.

RESPONSE

Following the guidelines, we have added and updated the captions and citations of the supporting information file at the end of the manuscript.

 

Editor

Thank you for your valuable feedback and suggestions. Please find below our point-by-point response to your comments and concerns.

We added the information about expenditures in 2020 from the Estimates of National Medical Care Expenditure, which were made available by the government on November 30th, 2022. As a result, we have updated the tables and figures, but these changes were only made to the results from 2020. We have not made any changes to the other results.

1. Rearrange the sequence of references. In P3, I see reference no.25 comes after citing the first 9 references.

RESPONSE

Sorry for our mistake. We have re-confirmed all the references and updated them.

2. P6: You stated that dental procedures are classified in 14 areas but based on what you wrote (A-O), they are supposed to be 15! Please check.

RESPONSE

We appreciate your comment regarding the category number, which is now correctly indicated as 15.

(Line 106–107, page 6)

The MHLW classified dental procedures into 15 areas:

3. In the reference list, kindly update the access date after checking their availability.

RESPONSE

We have updated and confirmed the references and their availability.

 

Reviewer 1

We appreciate the time and effort that you have invested in reviewing our manuscript. We have responded to all comments carefully, as shown below.

We added the information about expenditures in 2020 from the Estimates of National Medical Care Expenditure, which were made available by the government on November 30th, 2022. As a result, we have updated the tables and figures, but these changes were only made to the results from 2020. We have not made any changes to the other results.

1. Abstract: medical management category should be explicit.

RESPONSE

We have added the explanation of "medical management" category in the abstract.

(Line 18–20, page 2)

In 0-14 years persons, expenses on the crown restoration and prosthesis category decreased while the medical management category (mainly including fees for a management plan for oral diseases or oral functions) increased.

2. Abstract: The conclusion section should have the main results in quantitative statements as well.

RESPONSE

We have revised the conclusion section based on the quantitative results.

(Line 23–26, page 2–3)

Conclusion

The amount of dental spending in Japan substantially increased from 1.96 trillion yen in 1984 to 3.00 trillion yen in 2020), a 1.53-fold increase. The observed changes in annual dental spending varied across age groups and types of service.

3. Page 4, Line 58: You have stated the proportion of older persons who had 20 teeth or more in 1993 and 2016. It is too old. Authors need to report the latest data.

RESPONSE

We agree that this information is old. However, due to the pandemic of the COVID-19, the government cancelled the Survey of Dental Diseases that was scheduled for 2021, and as a result, the most recent available national data is from 2016 because the government conducts the survey every five years. Furthermore, there are no national surveys that report the oral health status of the entire Japanese population. Only the DMFT index of 12 years old is available because another national survey collects the health status of schoolchildren.

4. This study design did not control or adjust for underlying secular trends, so it may have failed to adequately capture dental expenditure trends. Moreover, it was not able to minimize threats to internal validity, including random fluctuations, history, and maturation; these extraneous events and changes over time.

RESPONSE

We agree with you. Although the Estimates of National Medical Care Expenditure covers all the expenditures of insurance services in Japan, as this descriptive study did not consider potentially underlying secular trends, the trends we found in this study might be biased. Therefore, we have toned down the expressions of the discussion and stated the limitations in the discussion section.

(Line 204–209, page 11)

First, the datasets of SECHC were June data and not annual data [23]. Therefore, the proportions of each type of dental service have random errors. Furthermore, as the sampling method has changed since 2015 [23], the data before 2014 have potentially increased the random errors. Moreover, this study did not account for underlying secular trends and extraneous factors. Therefore, these limitations can violate internal validity.

5. Your references and literature search shows that studies have been conducted on the trend in dental expenditures under the universal health insurance system in Japan. Please highlight what additional value your study adds and the justification for your study.

RESPONSE

We have emphasized the novelty of this study compared to previous studies.

(Line 37–42, page 3)

Previous studies on dental expenditures in Japan have primarily focused on total costs without examining the spending pattern across different age groups, types of services, and over time. A detailed examination of trends in dental expenditures is crucial for the development of future sustainable plans for the allocation of financial resources. By understanding the specific patterns and trends in dental expenditures, policymakers can make decisions to ensure effective distribution of resources in the dental healthcare sector.

---

## [Editor Report · Decision Letter 1]

12 Sep 2023

PONE-D-22-32885R1Trends in dental expenditures in Japan with a universal health insurance systemPLOS ONE

Dear Dr. Sato,

Thank you for submitting your manuscript to PLOS ONE. After careful consideration, we feel that it has merit but does not fully meet PLOS ONE’s publication criteria as it currently stands. Therefore, we invite you to submit a revised version of the manuscript that addresses the points raised during the review process.

We look forward to receiving your revised manuscript.

Kind regards,

Ahmed Jamleh

Academic Editor

PLOS ONE

Journal Requirements:

Additional Editor Comments:

L37-39: Please consider adding references. With this, you need to consider rearranging the references

---

## [Author Response · Author response to Decision Letter 1]

20 Sep 2023

RESPONSES TO REVIEWERS’ COMMENTS

Dear Professor Ahmed Jamleh,

We would like to thank you for your insightful comment on our paper. We have responded to the comment and made changes to the manuscript accordingly.

Journal requirements

RESPONSE

Thank you for checking the references in our manuscript. We have confirmed that there was no retracted reference. We have made minor revisions to the citation information. In addition, following the editor's comment, a new reference has been added.

"2. Sakamoto H, Rahman M, Nomura S, Okamoto E, Koike S, Yasunaga H, et al. Japan health system review. New Delhi: World Health Organization. Regional Office for South-East Asia; 2018. Available: https://apps.who.int/iris/handle/10665/259941"

was revised to

"2. Sakamoto H, Rahman M, Nomura S, Okamoto E, Koike S, Yasunaga H, et al. Japan health system review. Health Syst Transit. 2018;8. Available: https://apps.who.int/iris/handle/10665/259941"

"5. OECD. Health at a Glance 2021. 2021. doi:https://doi.org/https://doi.org/10.1787/ae3016b9-en"

was revised to

"5. OECD. Health at a Glance 2021. 2021. Available: https://www.oecd-ilibrary.org/content/publication/ae3016b9-en"

"7. OECD. Demography: Population. In: OECD Data [Internet]. [cited 17 Nov 2022]. Available: http://data.oecd.org/pop/population.htm"

was changed to 

"7. Statistics Bureau of Japan. Population estimates in Japan (in Japanese). 2022 [cited 25 Feb 2022]. Available: https://www.stat.go.jp/data/jinsui/2.html"

"9. Brennan D. Dental Health Services Epidemiology. In: Peres MA, Antunes JLF, Watt RG, editors. Oral Epidemiology: A Textbook on Oral Health Conditions, Research Topics and Methods. Cham: Springer International Publishing; 2021. pp. 395–407. doi:10.1007/978-3-030-50123-5_26"

was added.

"9. Japan Health Policy NOW. Japanese Health Policy. [cited 17 Nov 2022]. Available: http://japanhpn.org/en/_japan_healthpolicy/"

was revised to

"11. Health and Global Policy Institute. Japanese Health Policy. 2019 [cited 17 Nov 2022]. Available: http://japanhpn.org/en/_japan_healthpolicy/"

"13. Kamijo H. Reform of the Social Security System and Prospects for Oral Health and Care in Japan -Influence of daily dental treatment and management of dental clinics- (in Japanese). SHIKWA GAKUHO. 2017;117: 1–16. doi:10.15041/tdcgakuho.117.1"

was revised to

"15. Kamijo H. Reform of the Social Security System and Prospects for Oral Health and Care in Japan -Influence of daily dental treatment and management of dental clinics- (in Japanese). J Tokyo Dent Coll Soc. 2017;117: 1–16. doi:10.15041/tdcgakuho.117.1"

"20. Nakai M, Niinomi M. Chapter 12 Dental Metallic Materials. Advances in Metallic Biomaterials: Processing and Applications. Springer; 2015."

was revised to

"20. Nakai M, Niinomi M. Dental Metallic Materials. In: Niinomi M, Narushima T, Nakai M, editors. Advances in Metallic Biomaterials. Berlin, Heidelberg: Springer; 2015. pp. 251–281. doi:10.1007/978-3-662-46842-5_12

"24. Statistics Bureau, Ministry of Internal Affairs and Communications. Consumer Price Index (in Japanese). In: Consumer Price Index [Internet]. [cited 14 Jun 2022]. Available: https://www.stat.go.jp/data/cpi/"

was revised to 

"24. Statistics Bureau, Ministry of Internal Affairs and Communications. Consumer Price Index (in Japanese). [cited 20 Sep 2023]. Available: https://www.stat.go.jp/data/cpi/"

 

Editor

1. L37-39: Please consider adding references. With this, you need to consider rearranging the references

RESPONSE

We wish to thank the editor for this comment. We have added references; thus, all the references have been rearranged.

(Line 37–43, page 3)

Previous studies on dental expenditures in Japan have primarily focused on total costs without examining the spending pattern across different age groups, types of services, and over time [3,8]. A detailed examination of trends in dental expenditures is crucial for the development of future sustainable plans for the allocation of financial resources [9]. By understanding the specific patterns and trends in dental expenditures, policymakers can make decisions to ensure effective distribution of resources in the dental healthcare sector [9,10].

---

## [Editor Report · Decision Letter 2]

25 Sep 2023

Trends in dental expenditures in Japan with a universal health insurance system

PONE-D-22-32885R2

Dear Dr. Sato,

We’re pleased to inform you that your manuscript has been judged scientifically suitable for publication and will be formally accepted for publication once it meets all outstanding technical requirements.

Kind regards,

Ahmed Jamleh

Academic Editor

PLOS ONE
---

## [Editor Report · Acceptance letter]

27 Sep 2023

PONE-D-22-32885R2 

Trends in dental expenditures in Japan with a universal health insurance system 

Dear Dr. Sato:

I'm pleased to inform you that your manuscript has been deemed suitable for publication in PLOS ONE. Congratulations! Your manuscript is now with our production department. 

Kind regards, 

on behalf of

Professor Ahmed Jamleh 

Academic Editor

PLOS ONE